# Cross-modal representation of chewing food in posterior parietal and visual cortex

Tomohiro Ishii[1], Noriyuki Narita[2]*, Sunao Iwaki[3], Kazunobu Kamiya[2], Michiharu Shimosaka[4], Hidenori Yamaguchi[4], Takeshi Uchida[5], Ikuo Kantake[5], Koh Shibutani[4]

1 Department of Removable Prosthodontics and Geriatric Oral Health, Nihon University School of Dentistry at Matsudo, Matsudo, Chiba, Japan, 2 Research Institute of Oral Science, Nihon University School of Dentistry at Matsudo, Matsudo, Chiba, Japan, 3 Human Informatics and Interaction Research Institute, National Institute of Advanced Industrial Science and Technology (AIST), Tsukuba, Ibaraki, Japan, 4 Department of Anesthesiology, Nihon University School of Dentistry at Matsudo, Matsudo, Chiba, Japan, 5 Dental Support Co. Ltd., Chiba, Chiba, Japan

* narita.noriyuki@nihon-u.ac.jp

**Data Availability Statement:** Data are available via https://doi.org/10.5061/dryad.p5hqbzkz6. The data is available at this link and is ready for download.

## Abstract

Even though the oral cavity is not visible, food chewing can be performed without damaging the tongue, oral mucosa, or other intraoral parts, with cross-modal perception of chewing possibly critical for appropriate recognition of its performance. This study was conducted to clarify the relationship of chewing food cross-modal perception with cortex activities based on examinations of the posterior parietal cortex (PPC) and visual cortex during chewing in comparison with sham chewing without food, imaginary chewing, and rest using functional near-infrared spectroscopy. Additionally, the effects of a deafferent tongue dorsum on PPC/visual cortex activities during chewing performance were examined. The results showed that chewing food increased activity in the PPC/visual cortex as compared with imaginary chewing, sham chewing without food, and rest. Nevertheless, those activities were not significantly different during imaginary chewing or sham chewing without food as compared with rest. Moreover, subjects with a deafferent tongue dorsum showed reduced PPC/visual cortex activities during chewing food performance. These findings suggest that chewing of food involves cross-modal recognition, while an oral somatosensory deficit may modulate such cross-modal activities.

## Introduction

Previous research has confirmed visual cortex activation in the human brain during chewing [1, 2], though the functional significance of the visual cortex while performing food chewing has not been fully elucidated.

The use of braille was found to induce visual cortex activity in blind subjects [3–5], while experimental vision loss in healthy subjects resulted in cross-modal visual cortex activation during Braille reading [6–8]. In consideration of results showing cross-modal somatosensory and visual cortex associations during finger discrimination in blind and blindfolded healthy

**Funding:** This work was supported by the Dental Support Co., Ltd. and a JSPS KAKENHI (Grant No. 25463029). TU and IK receives a salary by the Dental Support Co., Ltd.. There was no additional external funding received for this study. The funders were not involved in any way in the study design, data collection and analysis, decision to publish, or preparation of the manuscript.

**Competing interests:** The remaining authors declare that the research was conducted in the absence of any commercial or financial relationships that could be construed as a potential conflict of interest. Funders provided support in the form of salaries for authors, but did not have any additional role in the study design, data collection and analysis, decision to publish, or preparation of the manuscript. The specific roles of these authors are articulated in the 'author contributions' section. This does not alter our adherence to PLOS ONE policies on sharing data and materials.

subjects [3–8], it is speculated that oral food perception during chewing food performance may have cross-modal communication with visual cortex activity in the blinded oral cavity. Thus, cross-modal somatosensory and visual association may be advantageous for spatial recognition and memory related to chewing food [9–14].

We previously investigated PPC/visual cortex activation during oral shape discrimination as compared with sham task performance, and those findings suggested a cross-modal association between oral somatosensory and PPC/visual cortex activities during oral tactual shape discrimination [15]. Should a relationship of oral register discrimination with PPC/visual cortex activities exist, then masticatory function for forming a food mass without damaging the oral cavity during mastication may also require cross-modal communication of oral somatosensory and visual sensations, with those activities shown during mastication. Based on speculation that PPC/visual cortex activities are exhibited during mastication, the present study was conducted to further investigate the association between oral somatosensory and those activities during chewing food performance by contrasting chewing food with sham chewing without food, imaginary chewing, and rest conditions. Furthermore, examinations of the effects of blunting of food percepts with tongue dorsal anesthesia on PPC/visual cortex activities during chewing performance were also performed.

We previously reported an overview of oral somatosensory and visual associations during chewing at an international dental meeting sponsored by the International Association for Dental, Oral, and Craniofacial Research [16]. Presented here are detailed findings of cross-modal food percepts related to PPC/visual cortex activities induced by a deafferent tongue, not only during physiological but also pathological chewing. It is considered that findings indicating cross-modality of masticatory food masses would be helpful for clarification regarding the direct relationship between tactile and visual cross-modal food recognition of those food masses, as well as masticatory performance, in addition to the previously reported relationship of oral shape perception with masticatory performance. Functional near-infrared spectroscopy (fNIRS) has been utilized to examine PPC/visual cortex activities [15, 17–20]. In the present study, fNIRS was applied for several reasons, including 1) its previous use for evaluations of those activities in regard to oral discrimination, 2) availability for examination of mastication tasks in a normal setting, and 3) ease of establishing the experimental environment and use by the subjects. In the present study, fNIRS was employed to investigate manifestations of cross-modal associations of oral somatosensory-PPC/visual cortex activities during food chewing. This novel study then used those results to reveal the effects of chewing food perception on PPC/visual cortex activities during chewing performance. Additionally, discussion from the viewpoint of cross-modal food recognition while chewing is presented.

## Materials and methods

### Subjects

Nine right-handed healthy adult Asian males aged 23 to 47 years (30.8 ± 8.8 years, mean ± SD) participated in this study, with hand dominance confirmed using the Edinburgh Handedness Inventory [21]. All subjects were staff members of Nihon University School of Dentistry at Matsudo. Sensitivity power analysis was conducted to confirm the validity of the sample size in this study. Those findings indicated that results of an F-test obtained with the G*power software package [22] (alpha = 0.05, power = 0.80, number of measurements = 5) and with the present number of samples would be sufficiently powerful to detect a minimum effect size of f = 0.391. Inclusion criteria included the following: 1) no symptoms indicating temporomandibular joint or masticatory muscle dysfunction in examinations conducted using diagnostic criteria for temporomandibular disorders [23]; 2) mentally healthy, as indicated by a score of

<7 on the Hospital Anxiety and Depression Scale [24]; 3) no abnormalities found in a cranial examination; and 4) not currently taking any medication. Exclusion criteria for the subjects included: 1) presence of temporomandibular joint or masticatory muscle dysfunction; 2) mental impairment, as indicated by a score of $\geq 7$ on the Hospital Anxiety and Depression Scale [24]; and 3) presently taking medication. Written informed consent was obtained from each enrolled subject and the study was approved by the Ethics Committee of Nihon University School of Dentistry at Matsudo (EC 14–015 and EC19-14-015-1). This study was conducted in accordance with the provisions of the 1975 Declaration of Helsinki, revised in 2013.

## Experimental procedures

The experiments were performed in a quiet room, with a screen positioned around the subject to block extra visual information from entering the field of view and affecting visual perception. The subject was seated in a chair during the trial. Each received an explanation regarding the experimental procedures, which were conducted using fNIRS with probes attached to the back of the head. After the explanation, the probes were attached to the parietal and occipital lobe regions. Cerebral blood flow was measured using fNIRS during five different sessions, as follows: 1) rest during the task period (Rest), 2) chewing gum (Chewing), 3) imaginary chewing (Image), 4) sham chewing without gum (Sham), and 5) chewing with tongue surface anesthesia (Anesthesia) in each subject. The Rest, Chewing, Image, and Sham sessions were conducted in a random manner to prevent effects caused by the order of the sessions, while the Anesthesia session was performed last with all of the subjects. After finishing all five sessions, the three-dimensional locations of each probe and the landmark position (NZ, Iz, AI, A2, Cz) on the scalp were recorded. The experiments were conducted and data collected between July 20, 2014 and December 10, 2014.

## Sessions and task trials

fNIRS data were collected during five sessions with the subject's eyes open. Each session was composed of five different task performances or trials, as follows: 1) Rest, 2) Chewing, 3) Image, 4) Sham, and 5) Anesthesia. Each session was started with a pre-trial rest period of 40 seconds, then repeated five times for each trial period conducted for 50 seconds, and finished with a post-trial period of 40 seconds. One trial consisted of a 20-second pre-task period, a 10-second task period, and a 20-second post-task period, with each repeated five times during one session. The total measurement time for one session was 330 seconds (Fig 1A), with a five-minute interval between each session.

For the chewing tasks, the subject was asked to chew, imagine chewing, or simulate chewing gum on both sides equally for a period of 10 seconds. A single piece of chewing gum (1 g, hardness $2.3 \times 10^5$ Pa-S, tasteless, Lotte Co., Japan) was used as the test food as appropriate for the task. The examiner instructed each subject to remain quiet until given a verbal cue. After the examiner said "start", the subject was asked to begin to chew the gum, imagine chewing gum, or simulate chewing gum until the examiner said "stop". They were also instructed to avoid head movements during performance of the task.

## Anesthesia

For the Anesthesia session, 0.2 g of Neozalocain paste (Neo Dental International, Inc, USA) was placed on the anterior two-thirds of the tongue dorsal surface, which provides sensory innervation by the lingual nerve, with a cotton swab for five minutes in order to investigate the effects of deprivation of sensory input from the tongue. The effects of anesthesia were assessed both objectively and subjectively. For the subjective assessment, the subject was asked about

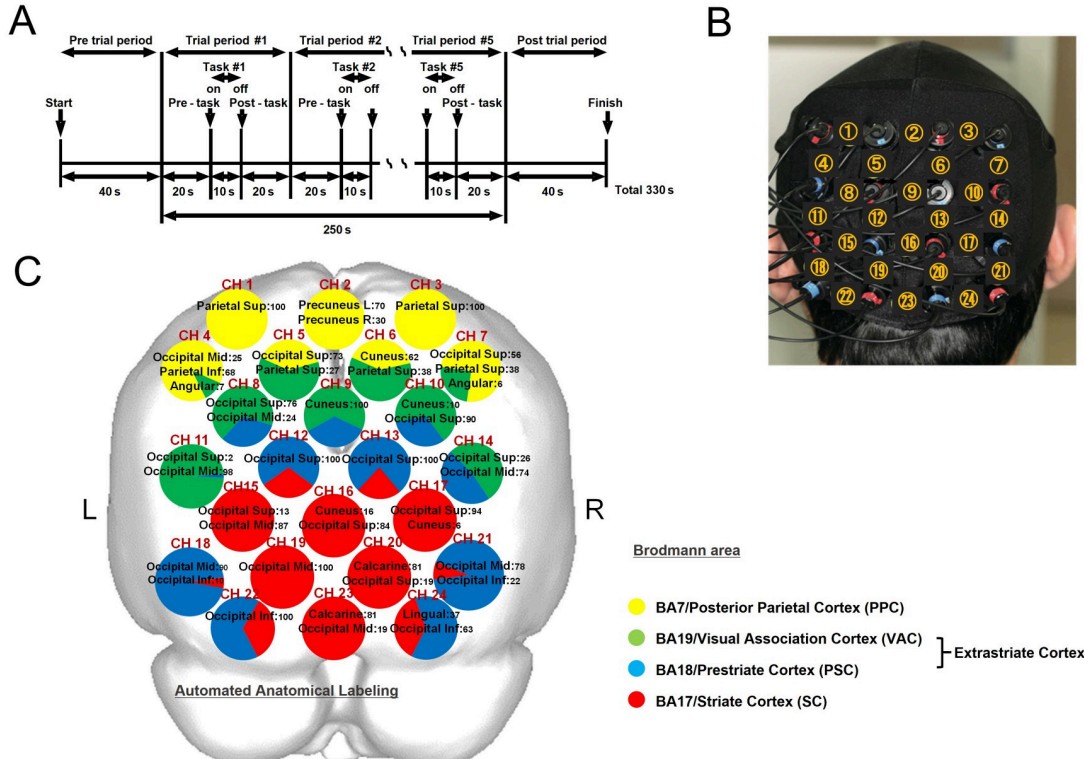

**Fig 1. Experimental design, position of fNIRS probes, and anatomical identification of fNIRS channels.** A. Each session started with a pre-trial period of 40 seconds, then was repeated five times for each trial period of 50 seconds, and finished with a post-trial period of 40 seconds. One trial consisted of a 20 second pre-task period, a 10 second task period, and a 20 second post-task period. Each trial was repeated five times. The total measurement time for one session was 330 seconds. B. Each probe was fitted with a 4 × 4 thermoplastic shell and placed at the center of the bottom line of shell, which was positioned on the inion (Iz) according to the international 10–20 system. The distance between the tip of the probe and bottom of the shell was 10 mm. C. The coordinates for all probe and anatomical landmark positions (Nz, Iz, A1, A2, Cz) were obtained using a three-dimensional digitizer. Yellow indicates the posterior parietal cortex (PPC) [Brodmann area (BA) 7], green the visual association cortex (VAC) (BA19), blue the prestriate cortex (PSC) (BA18), and red the striate cortex (SC) (BA17). Each circle corresponds to a channel and the pie chart within each circle shows the percentage of areas in that channel.

sense of numbness on the tongue [25]. For the objective assessment, loss of sensation in six regions of the dorsal tongue was determined using a pinprick test, during which a sharp dental probe was applied to the mucosa and moved in a pricking manner [26]. Pinprick stimulation has been shown to predominantly activate Aδ fibers [27].

Numbness on the tongue dorsal surface was assessed before and after anesthesia with a visual analogue scale (VAS), a 100 mm horizontal line with the left end representing no numbness and right end representing the worst imaginable numbness. Numbness is often evaluated using subjective methods such as the VAS. The anesthesia session was performed last for all subjects.

## fNIRS measurements

fNIRS signals were recorded using a 24-channel fNIRS device (ETG-100, Hitachi Medical Co., Chiba, Japan), which utilizes near-infrared light at two wavelengths, 780 and 830 nm [28, 29]. The distance between each pair of detector probes was 30 mm and the device was set to measure at points associated with the surface of the cerebral cortices [30, 31]. The probes were fitted with 4 × 4 thermoplastic shells and placed at the center of the bottom line of the shell,

which was positioned on the inion (Iz) according to the international 10–20 system [32–34]. The distance between the tip of the probe and bottom of the shell was 10 mm (Fig 1B).

## Anatomical localization of fNIRS channels

A 3-D magnetic spatial digitizer (3SPACE ISOTRACK2, Polhemus, USA) was used to record the position of each probe and landmark positions (NZ, Iz, AI, A2, Cz) in three dimensions on the scalp of each participant. Furthermore, estimation of the corresponding location of each channel in the Montreal Neurological Institute (MNI) space [35, 36] was obtained with use of a probabilistic registration method [37, 38], with anatomical localization corresponding to the probe position identified using the Platform for Optical Topography Analysis Tools (POTATo) (Adv. Res. Lab., Hitachi Ltd. Japan), with reference to Automated Anatomical Labeling [39, 40]. POTATo, a plug-in-based analysis platform that runs on MATLAB (The MathWorks Inc. USA) [38, 41], processes concentration changes of oxygenated-hemoglobin ([oxy-Hb]), deoxygenated-hemoglobin ([deoxy-Hb]), and total hemoglobin ([total-Hb]) using differential absorption proportional values of near infrared light detected by an fNIRS device [42, 43], then converts each channel position into a normalized brain surface [44, 45].

Anatomical identification of the NIRS channels is shown in Fig 1C. Twenty-four measurement channels were divided into five regions of interest (ROIs) [46–48], including the posterior parietal cortex (PPC), posterior parietal cortex/visual association cortex (PPC/VAC), VAC/prestriate cortex (VAC/PSC), PSC/striate cortex (PSC/SC), and SC. ROI-1 (channels 1, 2, 3) was located in the PPC [Brodmann area (BA) 7], ROI-2 (channels 4, 5, 6, 7) in the PPC (BA 7)/VAC (BA19), ROI-3 (channels 8, 9, 10, 11, 14) in the VAC (BA19)/PSC (BA18), ROI-4 (channels 12, 13, 18, 21, 22, 24) in the PSC (BA 18)/SC (BA17), and ROI-5 (channels 15, 16, 17, 19, 20, 23) in the SC (BA17). Subsequently, to reduce signal variations, the mean value for [oxy-Hb] for each ROI (averaged across channels) during the pre-task, task, and post-task periods was calculated for each experimental condition. (Fig 1C).

## fNIRS data analysis

fNIRS data were preprocessed for addition averaging using POTATo. Oxy-Hb signal results have been widely reported in clinical research studies [49], with findings indicating better sensitivity to task-related hemodynamic changes [50] and excellent reliability for task-related activities [51]. Based on our previous results [52–54], Oxy-Hb signals were focused on in the present study. The sampling interval was 0.1 seconds. Each trial was repeated five times and obtained values were averaged using the 'integral mode' of the ETG-100 software for the Rest, Chewing, Image, Sham, and Anesthesia sessions. Also, a linear fitting algorithm [55] was used for baseline corrections [56]. A moving average with a window width of 5 seconds was used to remove physiological noise such as cardiac artifacts [57] and short-term motion artifacts [58, 59] in the fNIRS signals.

## Statistical analysis

Numbness VAS scale results after anesthesia were compared with those before anesthesia using a one-sample signed rank test to objectively confirm the effects of anesthesia [60, 61]. The value for [oxy-Hb] was calculated every 1 second and compared between the Chewing and Rest, Image and Rest, Sham and Rest, and Anesthesia and Rest sessions using paired t-tests, implemented with a plug-in-based analysis platform that runs on MATLAB (The MathWorks Inc.), with values showing $p < 0.05$ considered to be significantly different. A topographical representation of significant channels every 1 second was projected onto the occipital cortical surface of a Montreal Neurological Institute standard brain space [62, 63]

using a three-dimensional composite display unit (version 2.41, Hitachi Medical Co. Chiba Japan) [64]. It has been shown that as the number of items being examined increases, so does the risk of type 1 errors [65]. Thus, in order to avoid such errors, two-way repeated measures ANOVA and multiple comparisons using a paired-t-test with Bonferroni correction were applied. The mean signal from the five ROIs was used for two-way repeated measures ANOVA.

Data for temporal changes of continuously averaged data for accumulated [oxy-Hb] every 1 section in each ROI did not support a normal distribution (Shapiro-Wilk, $p < 0.05$), thus nonparametric statistical analysis was performed using nonparametric two-way repeated measured analysis of variance with rank-transformed values (aligned ranked transformation, ART) [66]. ART is a modification of rank transformation [67, 68] that allows for accurate testing to determine interaction effects. By aligning the data to strip the interaction effect from the main effects, as well as the main effects from each other and the interaction, and then ranking it, mixed factorial ANOVA is possible [69]. Multiple comparisons using a paired t-test with Bonferroni correction were used as a post hoc test for comparisons of temporal changes of continuously averaged data of accumulated [oxy-Hb] for every 1 second in each ROI between the Chewing and Rest, Image and Rest, Sham and Rest, Anesthesia and Rest, Chewing and Sham, Chewing and Image, Chewing and Anesthesia, Image and Sham, Image and Anesthesia, and Sham and Anesthesia sessions. Effect size (f) values were accessed for ANOVA and also used to estimate the standardized degree of effect, including differences, and influence of the findings shown by the obtained data in this study. Eta squared ($\eta^2$) values were used to estimate effect size, with a magnitude for effect ($\eta^2$) of 0.01 defined as small, of 0.06 defined as medium, and of 0.14 defined as large [70]. The statistical software package SigmaPlot, ver. 14.5 (Systat Software, Inc., San Jose, CA, USA) was used for all analyses. *P*-values less than 0.05 were considered to indicate significance.

## Results

### Numbness on tongue dorsal surface before and after anesthesia

The numbness VAS scale value for all subjects before anesthesia was 0. A significant increase to 50.4 ± 0.4 (one-sample signed rank test, *p* = 0.004) was noted after starting anesthesia.

### Grand averaged waveforms and topographical maps

The topography of changes in [oxy-Hb] during the pre-task, task, and post-task periods in the Rest, Chewing, Image, Sham, and Anesthesia sessions is shown in Fig 2. Grand averaged waveforms for the nine subjects for changes in [oxy-Hb] and [deoxy-Hb] during the Rest, Chewing, Image, Sham, and Anesthesia sessions are presented in Fig 2A–2E, respectively. During the Rest session, there was no apparent change in [oxy-Hb] in the occipital lobe (Fig 2A). As for the Chewing session, there was an increase in [oxy-Hb] in the PPC (BA7), VAC (BA19), PSC (BA18), and SC (BA17) (Fig 2B), while in the Image session there was an increase in [oxy-Hb] in the SC (BA17) (Fig 2C) during the task period. Furthermore, in the Sham session, there was an increase in [oxy-Hb] in the PPC (BA7) and SC (BA17) (Fig 2D), and in the Anesthesia session there was an increase in [oxy-Hb] in the PPC (BA7), VAC (BA19), PSC (BA18), and SC (BA17) (Fig 2E) during the task period.

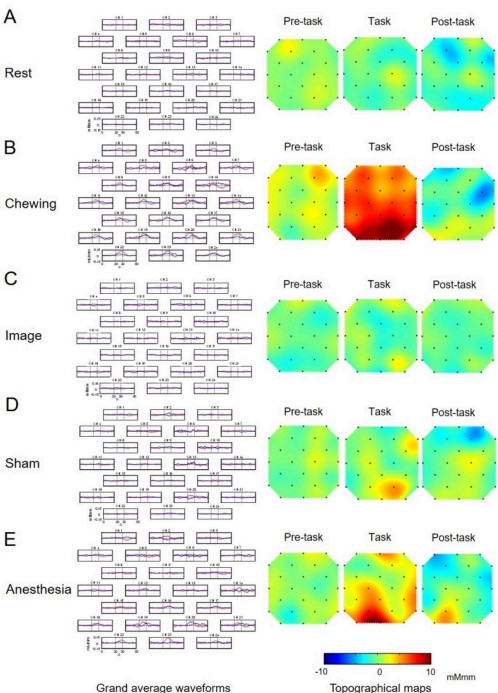

**Fig 2. Grand average waveforms and topographical maps.** Left: Grand average changes in oxygenated-hemoglobin concentration ([oxy-Hb], red line) and deoxygenated-hemoglobin concentration ([deoxy-Hb], blue line) during the Rest (A), Chewing (B), Image (C), Sham (D), and Anesthesia (E) sessions for each of the 24 measurement channels in the nine subjects. The x-axis indicates time (s) and y-axis hemodynamic change (mMmm). Grey vertical lines at 20 and 30 seconds indicate the start and end, respectively of the 10-second task period. Right: Topographical maps showing changes in [oxy-Hb] during the 10-second period preceding the task period (Pre-task), the 10-second task period (Task), and the 10-second period following the task period (Post-task) during the Rest (A), Chewing (B), Image (C), Sham (D), and Anesthesia (E) sessions. While there was no apparent change during Rest (A), there was a marked increase during the Chewing task period (B), a slight increase during the task period in the Image and Sham task periods (C, D), and a significant increase during the Anesthesia task and post-task periods (E).

## Temporal changes in [oxy-Hb] during rest, chewing, image, sham, and anesthesia

During the Rest, Chewing, Image, Sham, and Anesthesia sessions, parietal and occipital lobe activities showed significant interactions with time in the PPC (CH 1, 2, 3) [F = 1.399; $p < 0.001$, power = 0.943, $\eta^2 = 0.10$, f = 0.418, effect size larger than minimum (f = 0.391) based on sensitivity analysis], PPC/VAC (CH 4, 5, 6, 7) [F = 1.428, $p < 0.001$, power = 0.962, $\eta^2 = 0.11$, f = 0.423, effect size larger than minimum (f = 0.391) based on sensitivity analysis], VAC/PSC (CH 8, 9, 10, 11, 14) [F = 1.527, $p < 0.001$, power = 0.993, $\eta^2 = 0.11$, f = 0.437, effect size larger than minimum (f = 0.391) based on sensitivity analysis], PSC/SC (CH 12, 13, 18, 21, 22, 24) [F = 2.779, $p < 0.001$, power = 1.00, $\eta^2 = 0.16$, f = 0.539, effect size larger than minimum (f = 0.391) based on sensitivity analysis, and SC (CH 15, 16, 17, 19, 20, 23) [F = 3.312, $p < 0.001$, power = 1.00, $\eta^2 = 0.19$, p = 0.643, effect size larger than minimum (f = 0.391) based on sensitivity analysis].

**Comparison between chewing and rest.** When Chewing was compared to Rest, the values for [oxy-Hb] were significantly ($p < 0.05$) increased during the task periods in PPC (Fig 3A), PPC/VAC (Fig 3B), VAC/PSC (Fig 3C), PSC/SC (Fig 3D), and SC (Fig 3E).

**Comparison between image and rest.** The values for [oxy-Hb] were not significantly different between Image and Rest during the task periods.

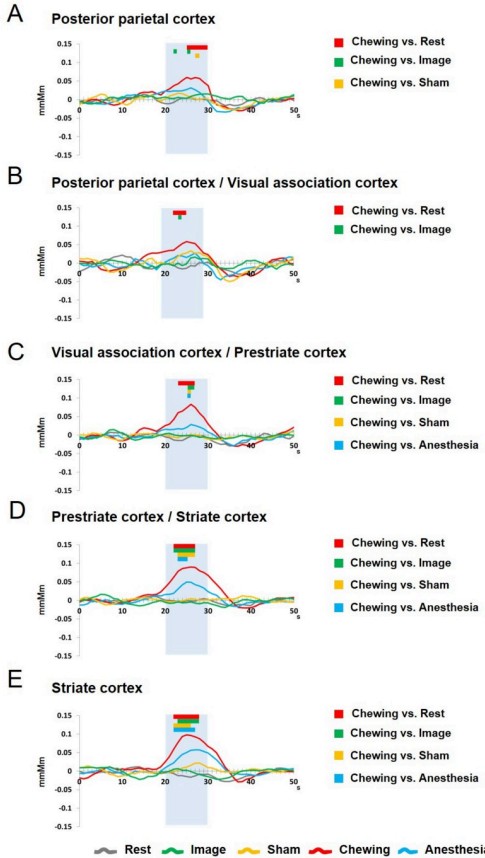

**Fig 3. Temporal changes in [oxy-Hb] during rest, chewing, image, sham, anesthesia.** Changes in [oxy-Hb] in the PPC, VAC, PSC, and SC during the Rest, Chewing, Image, Sham, and Anesthesia sessions are shown. Significant differences were clearly present between Chewing and Rest (red bar) in the PPC, PPC/VAC, VAC/PSC, PSC/SC, and SC; between Chewing and Image (green bar) in the PPC, PPC/VAC, VAC/PSC, PSC/SC, and SC; between Chewing and Sham (orange bar) in the PPC, VAC/PSC, PSC/SC, and SC; and between Chewing and Anesthesia (blue bar) in the VAC/PSC, PSC/SC, and SC. Light blue shading indicates the 10-second task period.

**Comparison between sham and rest.** The values for [oxy-Hb] were not significantly different between Sham and Rest during the task periods.

**Comparison between anesthesia and rest.** The values for [oxy-Hb] were not significantly different between Anesthesia and Rest during the task periods.

**Comparison between chewing and image.** When Chewing was compared to Image, the values for [oxy-Hb] were significantly ($p < 0.05$) increased during the task periods in PPC (Fig 3A), PPC/VAC (Fig 3B), VAC/PSC (Fig 3C), PSC/SC (Fig 3D), and SC (Fig 3E).

**Comparison between chewing and sham.** When Chewing was compared to Sham, the values for [oxy-Hb] were significantly ($p < 0.05$) increased during the task periods in PPC (Fig 3A), VAC/PSC (Fig 3B), PSC/SC (Fig 3D), and SC (Fig 3E).

**Comparison between chewing and anesthesia.** When Chewing was compared to Anesthesia, the values for [oxy-Hb] were significantly ($p < 0.05$) increased during the task periods in VAC/PSC (Fig 3C), PSC/SC (Fig 3D), and SC (Fig 3E).

## Discussion

### Visual cortex activities during chewing, imaginary chewing, and sham chewing without food, and during rest

This study was performed as an investigation of parietal and occipital cortex activities during chewing performance. Significant increases in PPC/visual cortex activities were found while chewing food as compared with imaginary chewing, sham chewing without food, and rest settings, whereas there were no significant differences in those activities when imaginary chewing and sham chewing without food were compared with rest.

Previous studies have reported a cross-modal association of finger somatosensory with visual cortex activities during Braille reading in blind subjects [3–5], while blindfolded healthy subjects have also been found to have induced visual cortex activities during Braille reading [6–8]. Kagawa et al. [15] presented findings regarding oral somatosensory and visual cross-modality that showed PPC/visual cortex activities during oral shape tactual discrimination when visual receptivity was eliminated. Moreover, Ptito et al. [71], Kupers et al. [72], and Vuillerme et al. [73] each reported details related to visual cortex activities induced by tongue dorsal electrical stimulation in healthy controls. Based on these findings, it is considered that indications of PPC/visual cortex activities during chewing food performance suggest a cross-modal association of oral somatosensory with PPC/visual cortex activities involved in chewing food recognition during chewing performance in a non-visualized mouth. The present study also found no significant PPC/visual cortex activities during imaginary chewing or sham chewing without food, as compared to a rest condition, though it has been suggested the visual cortex activity occurs by means of mental imaginary performance [74–77]. Regarding the differences between these imaginary effects on PPC/visual cortex activities, it is considered that the vividity of the target might be different in mental imagery. Therefore, it is possible that the properties of the food bolus may be quite unclear during chewing in the invisible oral space [78–82]. In the present subjects, chewing food performance was shown to induce significant PPC/visual cortex activities as compared with the imaginary chewing, sham chewing, and rest conditions, thus suggesting a cross-modal oral somatosensory and PPC/visual cortex association while chewing food.

### Association of oral sensory deprivation with modulatory PPC/visual cortex activity during chewing performance

Results obtained in this study also show the effects of anesthetized oral sensory deprivation on reduced PPC/visual cortex activities during chewing performance. A decrease in those activities caused by oral anesthesia may also paradoxically indicate that cross-modal PPC/visual cortex activities are due to oral somatosensory food perception. Furthermore, such a pathological cross-modal communication between somatosensory deficits and decreased PPC/visual cortex activities suggests discriminative deficits in food recognition while chewing and an association with awareness of chewing disability, as seen in aged individuals with tooth loss [83–86]. It is also possible that such cross-modal effects may extend to multisensory chewing food recognition, such as the relationships of the somatosensory system with the taste cortex [87, 88] and auditory cortex [89, 90], in addition to the association of the somatosensory system with the PPC/visual cortex [91–93]. The present results showed that oral deprivation while chewing resulted in decreased PPC/visual cortex activity, which suggests that an oral somatosensory deficit can induce cross-modal effects on food recognition during chewing performance. It is thus speculated that PPC/visual cortex activities while chewing have a relationship with subjective chewing food ability in partially edentulous aged patients. Examinations of PPC/visual

cortex activities while chewing may be applicable for evaluation of oral food recognition ability for the field of dentistry.

## Study limitations and future research directions

In the present study, neural activity in the parietal and occipital lobes indicated a statistically significant interaction between task and time. Chewing food trials have found significant neural activation in the parietal and occipital lobes as compared to during rest, which were based on *p*-values and effect sizes used to evaluate the significance of the data [94]. Effect size represents the magnitude of change in outcome and is often more important than relying on *p*-value alone when interpreting study results [95]. Furthermore, effect size is independent of sample size [94]. Both *p*-value and effect size were larger than the minimum effect size value (0.391) obtained from sensitivity test power analysis in the present study, which is considered to confirm the significance of the results even though the sample size of 9 subjects is small. The results also confirmed that tactile-visual cross-modal substrates in the parietal and occipital cortex indicate shape perception in the oral cavity for chewing food perception during mastication, as also noted in a previous study [15]. Additionally, they indicate that oral sensory deprivation results in significantly decreased neural activity in the parietal and occipital lobes during food chewing. Thus, the requirement of representation of somatosensory-visual cross-modal food perception during chewing was confirmed. Nevertheless, in consideration of the variety of individual normal occlusion conditions encountered [96] as well as the influence of surface anesthesia in the mouth [97], the number of measurements in healthy subjects should be increased in future clinical trials in order to detect subtle differences between conditions, which may be a limitation of the present study.

Previous studies functionally evaluated chewing ability considered as on the particle size distribution in comminuted test food [98, 99], findings following glucose extraction from chewed food [100, 101], and color changes in chewing gum used for testing [102, 103]. In addition, the relationship [83, 84] of chewing efficiency with oral stereognosis ability has also been examined in young healthy and edentulous subjects. In future clinical studies, measurements of PPC/visual cortex activities during chewing can be applied to evaluate chewing ability from the viewpoints of tactile and visual cross-modal representations of chewing food in the dentistry patients with chewing difficulties.

## Conclusions

The functional associations of oral somatosensory food percepts with PPC/visual cortex activities during chewing were investigated in healthy subjects. Food chewing produced significant PPC/visual cortex activities as compared to imaginary chewing, sham chewing without food, and rest conditions. On the other hand, there were no significant differences between imaginary and sham chewing without food as compared with rest. In addition, oral somatosensory deprivation induced significant decreases in PPC/visual cortex activities during chewing, which suggests a pathological cross-modal representation of oral somatosensory-PPC/visual cortex activities. Based on these findings, it is considered that there are physiological and pathological cross-modal links between oral somatosensory and PPC/visual cortex activities in the process of chewing food recognition during chewing performance.

## Supporting information

**S1 File. Oxy-Hb raw data tables obtained under rest, chewing, sham, image, anesthesia condition.**
(XLSX)

## Author Contributions

**Conceptualization:** Noriyuki Narita.

**Formal analysis:** Tomohiro Ishii, Kazunobu Kamiya.

**Funding acquisition:** Ikuo Kantake.

**Investigation:** Tomohiro Ishii, Kazunobu Kamiya.

**Methodology:** Noriyuki Narita, Sunao Iwaki.

**Project administration:** Noriyuki Narita, Koh Shibutani.

**Resources:** Tomohiro Ishii, Kazunobu Kamiya, Michiharu Shimosaka, Hidenori Yamaguchi, Takeshi Uchida.

**Supervision:** Ikuo Kantake, Koh Shibutani.

**Validation:** Tomohiro Ishii, Kazunobu Kamiya, Michiharu Shimosaka, Hidenori Yamaguchi, Takeshi Uchida.

**Visualization:** Noriyuki Narita, Sunao Iwaki.

**Writing – original draft:** Tomohiro Ishii, Noriyuki Narita.

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
