## [Decision Letter · Decision Letter 0]

15 Feb 2024

PONE-D-23-39373Cross-modal representation of chewing food in posterior parietal and visual cortexPLOS ONE

Dear Dr. Narita,

Thank you for submitting your manuscript to PLOS ONE. After careful consideration, we feel that it has merit but does not fully meet PLOS ONE’s publication criteria as it currently stands. Therefore, we invite you to submit a revised version of the manuscript that addresses the points raised during the review process.

**As you will see below, and in accordance with my own assessment of the manuscript, both reviewers raise concerns about the small sample size and the correctness of the reported power analysis. I agree with the reviewers that not enough details are provided to replicate the power analysis in G*Power, but it appears unlikely that a sample size of n=3 would be sufficient to detect a medium-sized effect with 80% power. Please add sufficient details and double-check the correctness of the power calculation. In addition, I agree with Reviewer #2 that the small sample size of n=9 should be discussed as a limitation of the study. Please also address the additional points raised in the reviews, in particular make sure to add the methodological details recommended by Reviewer #2.**

We look forward to receiving your revised manuscript.

Kind regards,

Patrick Bruns

Academic Editor

PLOS ONE

Journal Requirements:

This work was supported by the Dental Support Co., Ltd. and a JSPS KAKENHI (Grant No. 25463029)

Takeshi Uchida and Ikuo Kantake are employed by Dental Support Co. Ltd. Neither has any potential conflicts of interest to declare with respect to research, authorship, and/or publication of this article. The remaining authors declare that the research was conducted in the absence of any commercial or financial relationships that could be construed as a potential conflict of interest.

We note that one or more of the authors are employed by a commercial company: Dental Support Co. Ltd

“The funder provided support in the form of salaries for authors, but did not have any additional role in the study design, data collection and analysis, decision to publish, or preparation of the manuscript. The specific roles of these authors are articulated in the ‘author contributions’ section.”

Reviewers' comments:

Reviewer's Responses to Questions

**Comments to the Author**

1. Is the manuscript technically sound, and do the data support the conclusions?

Reviewer #1: Yes

Reviewer #2: Yes

2. Has the statistical analysis been performed appropriately and rigorously? 

Reviewer #1: Yes

Reviewer #2: Yes

3. Have the authors made all data underlying the findings in their manuscript fully available?

Reviewer #1: Yes

Reviewer #2: Yes

4. Is the manuscript presented in an intelligible fashion and written in standard English?

Reviewer #1: Yes

Reviewer #2: Yes

5. Review Comments to the Author

Reviewer #1: The manuscript was easy to read and easy to understand. My main concern is with the sample size.

fNIRS should be mentioned in the title (+/-) abstract.

Methods: "Since the necessary minimum sample size was shown to be three..." Please report more details (options and parameters chosen in GPower 3) that led to this calculation. It looked strange, as very few neuroimaging papers nowadays advocate such a small sample size.

P11 L219 & P12 L237: "元文" should be removed.

Reviewer #2: Title: Cross-modal representation of chewing food in posterior parietal and visual cortex

The current study focuses on an interesting subject of cross-modal representation during food chewing. However, to enhance the manuscript, several modifications are recommended.

Introduction:

1. The current introduction is somewhat limited. To underscore the significance of the study, additional information about clinical implications should be incorporated.

2. I recommend including relevant literature using other neuroimaging tools and elucidating the rationale behind choosing fNIRS for the current investigation.

3. The inclusion of a clear statement of the hypothesis is recommended to provide a clear framework for the study.

Methods:

1. The variables considered for power analyses need to be explicitly stated for transparency.

2. Inclusion and exclusion criteria should be clearly outlined to enhance the study's transparency.

3. Demographic information about participants, including gender and ethnicity, should also be incorporated.

4. Did the author apply any methods to ensure consistent probe set placements across participants, for example, the use of facial landmarks?

5. Given that some studies found better correlations between fNIRS-related deoxygenated hemoglobin concentration and fMRI findings, it is advisable to reconsider the rationale for utilizing Oxy versus Deoxy values in the current study.

6. The data analysis pipeline should be explicitly described, encompassing filtering, removal of movement artifacts, and the model used to estimate fNIRS findings.

Results:

1. Sections of the description from lines 220-224 and lines 238-244 appear repetitive and should be revised for conciseness.

2. While significant time x ROI interactions were identified, the temporal distinctions across each ROI in post-hoc analyses were not specified.

Discussion:

1. It is essential to incorporate a discussion of study limitations, including the small sample size, and provide insights into potential future research directions.

6. PLOS authors have the option to publish the peer review history of their article (what does this mean?). If published, this will include your full peer review and any attached files.

Reviewer #1: No

Reviewer #2: **Yes: **Wan-Chun Su

---

## [Author Response · Author response to Decision Letter 0]

31 Mar 2024

Academic Editor

As you will see below, and in accordance with my own assessment of the manuscript, both reviewers raise concerns about the small sample size and the correctness of the reported power analysis. I agree with the reviewers that not enough details are provided to replicate the power analysis in G*Power, but it appears unlikely that a sample size of n=3 would be sufficient to detect a medium-sized effect with 80% power. Please add sufficient details and double-check the correctness of the power calculation. In addition, I agree with Reviewer #2 that the small sample size of n=9 should be discussed as a limitation of the study. Please also address the additional points raised in the reviews, in particular make sure to add the methodological details recommended by Reviewer #2.

Response

We appreciate the reviewer for pointing this out. In the original version, the number of measurements was incorrect. We were mistaken about the number of measurements. That has been corrected to 25 measurements (5 times repeated, 5 tasks) and the sample size was then recalculated using the following parameters: effect size = 0.3, α = 0.05, power (1-β) = 0.95, number of groups = 1, number of measurements = 25 (5 times repeated, 5 tasks). The sample size required was 9 and the power of this calculation was determined to be 0.974. The modified the Subjects section and the explanation regarding sample size has been added to the Discussion section in revised version.

Journal Requirements:

Response

We have verified that the manuscript meets PLOS ONE style requirements (including file naming).

This work was supported by the Dental Support Co., Ltd. and a JSPS KAKENHI (Grant No. 25463029)

Please provide an amended statement that declares all the funding or sources of support (whether external or internal to your organization) received during this study, as detailed online in our guide for authors at http://journals.plos.org/plosone/s/submit-now. Please also include the statement “There was no additional external funding received for this study.” in your updated Funding Statement. Please include your amended Funding Statement within your cover letter. We will change the online submission form on your behalf.

Response

The statement in the Financial Disclosure has been revised as follows.

Financial Disclosure

This work was supported by the Dental Support Co., Ltd. and a JSPS KAKENHI (Grant

No. 25463029). TU and IK receives a salary by the Dental Support Co., Ltd.. There was no additional external funding received for this study.

Takeshi Uchida and Ikuo Kantake are employed by Dental Support Co. Ltd. Neither has any potential conflicts of interest to declare with respect to research, authorship, and/or publication of this article. The remaining authors declare that the research was conducted in the absence of any commercial or financial relationships that could be construed as a potential conflict of interest.

We note that one or more of the authors are employed by a commercial company: Dental Support Co. Ltd

“The funder provided support in the form of salaries for authors, but did not have any additional role in the study design, data collection and analysis, decision to publish, or preparation of the manuscript. The specific roles of these authors are articulated in the ‘author contributions’ section.”

Response

The statement in the Funding Statement and Competing Interests Statement has been revised as follows.

Funding Statement and Competing Interests Statement 

This work was supported by the Dental Support Co., Ltd. and a JSPS KAKENHI (Grant

No. 25463029). Takeshi Uchida and Ikuo Kantake are employed by Dental Support Co. Ltd. Takeshi Uchida played roles in validation and resourcing for this study and Ikuo Kantake played roles in Funding acquisition for this study. The remaining authors declare that the research was conducted in the absence of any commercial or financial relationships that could be construed as a potential conflict of interest. Funders provided support in the form of salaries for authors, but did not have any additional role in the study design, data collection and analysis, decision to publish, or preparation of the manuscript. The specific roles of these authors are articulated in the ‘author contributions’ section. This does not alter our adherence to PLOS ONE policies on sharing data and materials.

Response

The authors confirm that all data underlying the findings described in their manuscript are fully available without restriction.

Reviewers' comments:

Reviewer's Responses to Questions

1. Is the manuscript technically sound, and do the data support the conclusions?

Reviewer #1: Yes

Reviewer #2: Yes

2. Has the statistical analysis been performed appropriately and rigorously? 

Reviewer #1: Yes

Reviewer #2: Yes

3. Have the authors made all data underlying the findings in their manuscript fully available?

Reviewer #1: Yes

Reviewer #2: Yes

4. Is the manuscript presented in an intelligible fashion and written in standard English?

Reviewer #1: Yes

Reviewer #2: Yes

5. Review Comments to the Author

Reviewer #1: The manuscript was easy to read and easy to understand. My main concern is with the sample size.

fNIRS should be mentioned in the title (+/-) abstract.

Response

Thank you for pointing this out. According to your suggestion, we added the “use of fNIRS” in the Abstract (p2 l8).

Even though the oral cavity is not visible, food chewing can be performed without damaging the tongue, oral mucosa, or other intraoral parts, with cross-modal perception of chewing possibly critical for appropriate recognition of its performance. This study was conducted to clarify the relationship of chewing food cross-modal perception with cortex activities based on examinations of the posterior parietal cortex (PPC) and visual cortex during chewing in comparison with sham chewing without food, imaginary chewing, and rest using functional near-infrared spectroscopy (fNIRS). Additionally, the effects of a deafferent tongue dorsum on PPC/visual cortex activities during chewing performance were examined. The results showed that chewing food increased activity in the PPC/visual cortex as compared with imaginary chewing, sham chewing without food, and rest, whereas that activity conducted with imaginary chewing and sham chewing without food was not significantly different as compared with rest. Moreover, subjects with a deafferent tongue dorsum showed reduced PPC/visual cortex activities during chewing food performance. Together, these results suggest that representation of PPC/visual cortex activities while chewing food suggests cross-modal recognition of chewing, while an oral somatosensory deficit may modulate cross-modal PPC/visual cortex activities during chewing performance.

Methods: "Since the necessary minimum sample size was shown to be three..." Please report more details (options and parameters chosen in GPower 3) that led to this calculation. It looked strange, as very few neuroimaging papers nowadays advocate such a small sample size.

Response

We appreciate the reviewer for pointing this out. In the original version, the number of measurements was incorrect. We were mistaken about the number of measurements. That has been corrected to 25 measurements (5 times repeated, 5 tasks) and the sample size was then recalculated using the following parameters: effect size = 0.3, α = 0.05, power (1-β) = 0.95, number of groups = 1, number of measurements = 25 (5 times repeated, 5 tasks). The sample size required was 9 and the power of this calculation was determined to be 0.974. The modified the section of the Subjects (p4 l64) and the explanation regarding sample size has been added to the Discussion section (p18 l372) in revised version.

(p4 l64)

The sample size was then recalculated using the following parameters: effect size = 0.3, α = 0.05, power (1-β) = 0.95, number of groups = 1, number of measurements = 25 (5 times repeated, 5 tasks). The sample size required was 9 and the power of this calculation was determined to be 0.974 [22]. 

(p18 l372)

Although, for the present study, two-way analysis of variance was used when both of the factors were corresponding. All factors were within-subject factors, which resulted in a greater number of measurements (25 times per subject). This within-subjects design has advantages, including fewer subjects required, minimized random noise, easier to find true differences between conditions, and increased power of the test [94-96]. The power calculation for nine participants is shown following: Power (1-β) = 0.974 [effect size = 0.3, power (β/α ratio) = 0.95, total sample size = 9, number of groups = 1, number of measurements = 25 (5 times repeated, 5 task)], and is considered to be sufficient. 

[94] Charness G, Gneezy U, Kuhn MA. Experimental methods: Between-subject and within-subject design. Journal of Economic Behavior & Organization. 2012; 81: 1–8. doi: 10.1016/j.jebo.2011.08.009

[95] Bellemare C, Bissonnette L, Kröger S. Statistical Power of Within and Between-Subjects Designs in Economic Experiments. Institute for Labor Economics (IZA), Discussion Paper no. 8583.

[96] Montoya AK. Selecting a Within- or Between-Subject Design for Mediation: Validity, Causality, and Statistical Power. Multivariate Behav Res. 2023; 58: 616-636. doi: 10.1080/00273171.2022.2077287.

P11 L219 & P12 L237: "元文" should be removed.

Response

Thanks for pointing out the mistake, we removed "元文".

Reviewer #2: Title: Cross-modal representation of chewing food in posterior parietal and visual cortex

The current study focuses on an interesting subject of cross-modal representation during food chewing. However, to enhance the manuscript, several modifications are recommended.

Introduction:

1. The current introduction is somewhat limited. To underscore the significance of the study, additional information about clinical implications should be incorporated.

Response

Thank you for pointing this out. We have included additional information regarding clinical implications in the Introduction section (p3 l32) in accordance with your suggestion.

If a relationship of oral register discrimination with PPC/visual cortex activities exists, then masticatory function for forming a food mass without damaging the oral cavity during mastication may also require cross-modal communication of oral somatosensory and visual sensations, resulting in those activities being shown during mastication. Based on speculation that PPC/visual cortex activities are exhibited during mastication, the present study was conducted to further investigate the association between oral somatosensory and those activities during chewing food performance by contrasting chewing food with sham chewing without food, imaginary chewing, and rest conditions.

2. I recommend including relevant literature using other neuroimaging tools and elucidating the rationale behind choosing fNIRS for the current investigation.

Response

Thank you for pointing this out. We have fo

---

## [Decision Letter · Decision Letter 1]

16 May 2024

PONE-D-23-39373R1Cross-modal representation of chewing food in posterior parietal and visual cortexPLOS ONE

Dear Dr. Narita,

Thank you for submitting your manuscript to PLOS ONE. After careful consideration, we feel that it has merit but does not fully meet PLOS ONE’s publication criteria as it currently stands. Therefore, we invite you to submit a revised version of the manuscript that addresses the points raised during the review process.

Your revised manuscript has been reviewed by original Reviewer #1 and myself. While the reviewer was happy with your changes and I agree that the review comments have been largely addressed, there appear to be some remaining issues with the power analysis which I have detailed in the Additional Editor Comments section below. As we try to avoid lengthy review processes with multiple rounds of revisions, please address these issues thoroughly with your next revision.

We look forward to receiving your revised manuscript.

Kind regards,

Patrick Bruns

Academic Editor

PLOS ONE

Journal Requirements:

Additional Editor Comments :

Power analysis: Could you please clarify how the parameters of the new power analysis map onto the statistical tests which are reported in the results section? Number of measurements was set to 25 (“5 times repeated, 5 tasks”): Does “5 times repeated” refer to the 5 trials per task condition which were recorded? It is my understanding that the 5 trials were averaged for the purpose of the statistical analyses (p. 10 l. 188), therefore trial repetition is not a repeated measurement factor in the analysis and the number of measurements for the main effect of condition would be 5 (i.e., the 5 task conditions) rather than 25. Note that G*Power does not seem to allow calculation of power for an interaction effect between two within-subject factors (such as Condition x Time Point), which might in fact require a simulation approach. For the main effect of condition, the reported power analysis in G*Power would yield a required sample size of n=22 to show an effect of size f=0.3 if the number of measurements were set to 5 (instead of 25). As an alternative to reporting an a priori power analysis, a sensitivity analysis could be reported, which would indicate that the sample size of n=9 had 95% power (with number of measurements set to 5) to show an effect of size f=0.49 (i.e., a large effect according to your definition on p. 12).

Other minor points

Abstract l. 15: Change “Together, these results suggest that representation of PPC/visual cortex activities while chewing food suggests…” to “Together, the result of PPC/visual cortex activities while chewing food suggests…”.

p. 4 l. 54-58: The last two sentences of the introduction appear somewhat redundant and could be condensed into one sentence.

p. 18 l. 367ff.: There appears to be some redundancy in this paragraph which could be streamlined.

Reviewers' comments:

Reviewer's Responses to Questions

**Comments to the Author**

1. If the authors have adequately addressed your comments raised in a previous round of review and you feel that this manuscript is now acceptable for publication, you may indicate that here to bypass the “Comments to the Author” section, enter your conflict of interest statement in the “Confidential to Editor” section, and submit your "Accept" recommendation.

Reviewer #1: All comments have been addressed

2. Is the manuscript technically sound, and do the data support the conclusions?

Reviewer #1: (No Response)

3. Has the statistical analysis been performed appropriately and rigorously? 

Reviewer #1: I Don't Know

4. Have the authors made all data underlying the findings in their manuscript fully available?

Reviewer #1: (No Response)

5. Is the manuscript presented in an intelligible fashion and written in standard English?

Reviewer #1: (No Response)

6. Review Comments to the Author

Reviewer #1: (No Response)

7. PLOS authors have the option to publish the peer review history of their article (what does this mean?). If published, this will include your full peer review and any attached files.

Reviewer #1: No

---

## [Author Response · Author response to Decision Letter 1]

16 Jul 2024

Response to Editor and Reviewers

Comment 1. Please review your reference list to ensure that it is complete and correct. Any changes to the reference list should be mentioned in the rebuttal letter that accompanies your revised manuscript.

Response

① Reference [22] was removed since reference [22] was used as the evidence for the sample size. Therefore, the subsequent reference numbers will be moved up by one and the total number of references will be changed to 103.

② Reference [69] was changed to the following reference because the Effect size was changed to f. We changed the reference from “69. Cohen, J. Statistical power analysis for the behavioral sciences. 2nd ed. Hillsdale, NJ: Lawrence Erlbaum; 1988”. to “69. Cohen, J. A power primer. Psychol Bull.1992; 112: 155-159. doi: 10.1037//0033-2909.112.1.155”.

③ The following three references were added to the Study limitation and future research directions section in the discussion.

93. Gail M. Sullivan, MD, MPH and Richard Feinn, PhD Using Effect Size—or Why the P Value Is Not Enough J Grad Med Educ. 2012 Sep; 4(3): 279–282. doi: 10.4300/JGME-D-12-00156.1

94. Julien KC, Buschang PH, Throckmorton GS, Dechow PC. Normal masticatory performance in young adults and children. Arch Oral Biol. 1996; 41: 69-75. doi: 10.1016/0003-9969(95)00098-4.

95. Månsson I, Sandberg N. Effects of surface anesthesia on deglutition in man. Laryngoscope. 1974; 84: 427-437. doi: 10.1288/00005537-197403000-00006.

Comment 2. Power analysis: Could you please clarify how the parameters of the new power analysis map onto the statistical tests which are reported in the results section? Number of measurements was set to 25 (“5 times repeated, 5 tasks”): Does “5 times repeated” refer to the 5 trials per task condition which were recorded? It is my understanding that the 5 trials were averaged for the purpose of the statistical analyses (p. 10 l. 188), therefore trial repetition is not a repeated measurement factor in the analysis and the number of measurements for the main effect of condition would be 5 (i.e., the 5 task conditions) rather than 25. Note that G*Power does not seem to allow calculation of power for an interaction effect between two within-subject factors (such as Condition x Time Point), which might in fact require a simulation approach. For the main effect of condition, the reported power analysis in G*Power would yield a required sample size of n=22 to show an effect of size f=0.3 if the number of measurements were set to 5 (instead of 25). As an alternative to reporting an a priori power analysis, a sensitivity analysis could be reported, which would indicate that the sample size of n=9 had 95% power (with number of measurements set to 5) to show an effect of size f=0.49 (i.e., a large effect according to your definition on p. 12).

Response

We sincerely appreciate your advice. According with your suggestion, we have changed to report the results of the sensitivity analysis calculation in the results section, instead of the sample size calculation. Accordingly, we have made the following changes.

① Removed sentence regarding sample size “The sample size was then recalculated using the following parameters: effect size = 0.3, α = 0.05, power (1-β) = 0.95, number of groups = 1, number of measurements = 25 (5 times repeated, 5 tasks). The sample size required was 9 and the power of this calculation was determined to be 0.974 [22]” in Materials and methods section.

② The sentence of effect size was changed from “Eta squared (η2) and r were used to estimate effect size. The effect size was used to estimate the standardized degree of effect, including differences, influence, correlations, and association of data. For effect size (η2), the magnitude of effect was defined as 0.01 = small, 0.06 = medium, and 0.14 = large, while for effect size (r), the magnitude of effect was defined as 0.1 = small, 0.3 = medium, and 0.5 = large [70].” to “Cohen’s effect size (f) values were accessed for ANOVA results. Effect size was used to estimate the standardized degree of effect, including differences, and influence of the findings shown by the obtained data. Critical Cohen's f values for distinguishing small, moderate, and large effects were 0.10, 0.25, and 0.40, respectively [69].” in Statistical analysis section,

③ The sentence of “The numbness VAS scale value for all subjects before anesthesia was 0. A significant increase to 50.4 ± 0.4 (one-sample signed rank test, p = 0.004, r = 0.89) was noted after starting anesthesia.” changed to “The numbness VAS scale value for all subjects before anesthesia was 0. A significant increase to 50.4 ± 0.4 (one-sample signed rank test, p = 0.004) was noted after starting anesthesia.” in Numbness on tongue dorsal surface before and after anesthesia section of results.

④ The sentence of “During the Rest, Chewing, Image, Sham, and Anesthesia sessions, parietal and occipital lobe activities showed significant interactions with time in the PPC (CH 1, 2, 3) (F = 1.399 p < 0.001, η2 = 0.10), PPC/VAC (CH 4, 5, 6, 7) (F = 1.428 p < 0.001, η2 = 0.11), VAC/PSC (CH 8, 9, 10, 11, 14) (F = 1.527 p < 0.001, η2 = 0.11), PSC/SC (CH , 12, 13, 18, 21, 22, 24) (F = 2.779, p < 0.001, η2 = 0.16), and SC (CH 15, 16, 17, 19, 20, 23) (F = 3.312, p < 0.001, η2 = 0.19).” changed to “During the Rest, Chewing, Image, Sham, and Anesthesia sessions, parietal and occipital lobe activities showed significant interactions with time in the PPC (CH 1, 2, 3) (F = 1.399 p < 0.001, power = 0.943, f = 0.482), PPC/VAC (CH 4, 5, 6, 7) (F = 1.428 p < 0.001, power = 0.962, f = 0.505), VAC/PSC (CH 8, 9, 10, 11, 14) (F = 1.527 p < 0.001, power = 0.993, f = 0.587), PSC/SC (CH , 12, 13, 18, 21, 22, 24) (F = 2.779, p < 0.001, power = 1.00, f = 0.662), and SC (CH 15, 16, 17, 19, 20, 23) (F = 3.312, p < 0.001, power = 1.00, f = 0.662).” in Temporal changes in [oxy-Hb] during Rest, Chewing, Image, Sham, and Anesthesia section of results,

Comment 3. Abstract l. 15: Change “Together, these results suggest that representation of PPC/visual cortex activities while chewing food suggests…” to “Together, the result of PPC/visual cortex activities while chewing food suggests…”.

Response

We sincerely appreciate your advice. According with your suggestion, we corrected “Together, these results suggest that representation of PPC/visual cortex activities while chewing food suggests…” to “Together, the result of PPC/visual cortex activities while chewing food suggests”.

Comment 4. p. 4 l. 54-58: The last two sentences of the introduction appear somewhat redundant and could be condensed into one sentence.

Response

We sincerely appreciate your advice. According with your suggestion, we changed “This novel study was conducted to reveal the effects of chewing food perception on PPC/visual cortex activities during chewing performance, with discussion from the viewpoint of cross-modal food recognition while chewing” to “This novel study then used those results to reveal the effects of chewing food perception on PPC/visual cortex activities during chewing performance. Additionally, discussion from the viewpoint of cross-modal food recognition while chewing is presented”.

5. p. 18 l. 367ff.: There appears to be some redundancy in this paragraph which could be streamlined.

Response

We sincerely appreciate your advice. According with your suggestion, we changed Study limitations and Future research directions section in Discussion to “In the present study, results showing neural activity in the parietal and occipital lobes indicated a statistically significant interaction between task and time. Chewing food trials have found significant neural activation in the parietal and occipital lobes as compared to resting task trials, based on obtained P-values and effect sizes [93]. Effect size represents the magnitude of change in an outcome and often can be more important than just relying on P-values when interpreting study results, as that indicates the actual magnitude of difference. On the other hand, the present findings indicate that oral sensory deprivation results in significantly decreased neural activity in the parietal and occipital lobes during food chewing. Thus, it is considered that the requirement of representation of somatosensory-visual cross-modality for food bolus perception during chewing was confirmed. Nevertheless, in consideration of the variety of individual normal occlusion conditions encountered [94] as well as the influence of surface anesthesia in the mouth [95], the number of measurements in healthy subjects should be increased in future clinical trials, which may be a limitation of the present study.

Previous studies functionally evaluated chewing ability based on particle size distribution in comminuted test food [96, 97], findings following glucose extraction from chewed food [98, 99], and color changes in chewing gum used for testing [100, 101]. In addition, the relationship [102, 103] of chewing efficiency with oral stereognosis ability has also been examined in young healthy and edentulous subjects. In future clinical studies, measurements of PPC/visual cortex activities during chewing can be applied to evaluate chewing ability from the viewpoints of tactile and visual cross-modal representations of chewing food in the dentistry patients with chewing difficulties.

---

## [Editor Report · Decision Letter 2]

23 Jul 2024

PONE-D-23-39373R2Cross-modal representation of chewing food in posterior parietal and visual cortexPLOS ONE

Dear Dr. Narita,

Thank you for submitting your manuscript to PLOS ONE. After careful consideration, we feel that it has merit but does not fully meet PLOS ONE’s publication criteria as it currently stands. Therefore, we invite you to submit a revised version of the manuscript that addresses the points raised during the review process.

As detailed below, I feel that the concerns regarding the power analysis have still not been fully addressed. Please do your utmost best to resolve these issues with your next revision.

We look forward to receiving your revised manuscript.

Kind regards,

Patrick Bruns

Academic Editor

PLOS ONE

Journal Requirements:

Additional Editor Comments:

Thank you for addressing my previous comments. However, I believe that the concerns regarding the power analysis have still not been fully resolved. It appears that what is now reported in the revised manuscript is a post-hoc power analysis, i.e. power was calculated based on the effect size that was observed in your study. Please note that reporting post-hoc power is usually considered pointless, as it does not add any information beyond the reported p values. Instead, a sensitivity power analysis should be performed as recommended in my comments on the previous version of the paper. A good discussion of the distinction between different types of power analyses and when to use them can, for example, be found in: Lakens (2022). Sample size justification. Collabra: Psychology 8(1). https://doi.org/10.1525/collabra.33267

On a related point: changing the reported effect sizes from eta squared to Cohen’s f values seems like a bad choice. Although both carry equivalent information and can be back-converted, eta squared is more commonly reported in ANOVAs and will thus be easier to interpret for readers.

---

## [Author Response · Author response to Decision Letter 2]

29 Aug 2024

Journal Requirements

Response

 Thank you for the suggestion. We have added the references, checked the citation order, and corrected the error.

Response to Editor

Comment 1.

 Thank you for addressing my previous comments. However, I believe that the concerns regarding the power analysis have still not been fully resolved. It appears that what is now reported in the revised manuscript is a post-hoc power analysis, i.e. power was calculated based on the effect size that was observed in your study. Please note that reporting post-hoc power is usually considered pointless, as it does not add any information beyond the reported p values. Instead, a sensitivity power analysis should be performed as recommended in my comments on the previous version of the paper. A good discussion of the distinction between different types of power analyses and when to use them can, for example, be found in: Lakens (2022). Sample size justification. Collabra: Psychology 8(1). https://doi.org/10.1525/collabra.33267

Response

 Thank you for your suggestion. I am very thankful that you even provided references. We have added a description of the sensitivity analysis in the subjects section (p4, l59). In addition, in the Results section, we have included an effect size for the 2-way ANOVA results and described the comparison to the minimum effect size obtained in the sensitivity analysis (p12, l234). In the Discussion section, in the Limitation section, we have modified the description of the validity of the 9 subjects and its limitations (p16, l329).

Comment 2. 

　　On a related point: changing the reported effect sizes from eta squared to Cohen’s f values seems like a bad choice. Although both carry equivalent information and can be back-converted, eta squared is more commonly reported in ANOVAs and will thus be easier to interpret for readers.

Response

 Thank you for your suggestion. We have combined η2 and f for the effect size in the description of the 2-way ANOVA results (p10, l194).

---

## [Editor Report · Decision Letter 3]

4 Sep 2024

Cross-modal representation of chewing food in posterior parietal and visual cortex

PONE-D-23-39373R3

Dear Dr. Narita,

We’re pleased to inform you that your manuscript has been judged scientifically suitable for publication and will be formally accepted for publication once it meets all outstanding technical requirements.

Kind regards,

Patrick Bruns

Academic Editor

PLOS ONE

Additional Editor Comments (optional):

Thank you for addressing all of my remaining comments.
---

## [Editor Report · Acceptance letter]

16 Oct 2024

PONE-D-23-39373R3 

PLOS ONE

Dear Dr. Narita, 

I'm pleased to inform you that your manuscript has been deemed suitable for publication in PLOS ONE. Congratulations! Your manuscript is now being handed over to our production team.

Kind regards, 

on behalf of

Dr. Patrick Bruns 

Academic Editor

PLOS ONE